# Bayesian Spatial Modeling of Anemia among Children under 5 Years in Guinea

**DOI:** 10.3390/ijerph18126447

**Published:** 2021-06-15

**Authors:** Thierno Souleymane Barry, Oscar Ngesa, Nelson Owuor Onyango, Henry Mwambi

**Affiliations:** 1Mathematics (Statistics Option) Program, Pan African University Institute for Basic Sciences, Technology and Innovation (PAUISTI), Nairobi 62000-00200, Kenya; 2Department of Mathematics and Physical Sciences, Taita Taveta University, Voi 635-80300, Kenya; oscanges@ttu.ac.ke; 3School of Mathematics, College of Biology and Physical Sciences, University of Nairobi, Nairobi 30197, Kenya; onyango@uonbi.ac.ke; 4School of Mathematics, Statistics and Computer Science, University of KwaZulu-Natal, Durban 4041, South Africa; MwambiH@ukzn.ac.za

**Keywords:** anemia, children under five, bayesian, spatial

## Abstract

Anemia is a major public health problem in Africa, affecting an increasing number of children under five years. Guinea is one of the most affected countries. In 2018, the prevalence rate in Guinea was 75% for children under five years. This study sought to identify the factors associated with anemia and to map spatial variation of anemia across the eight (8) regions in Guinea for children under five years, which can provide guidance for control programs for the reduction of the disease. Data from the Guinea Multiple Indicator Cluster Survey (MICS5) 2016 was used for this study. A total of 2609 children under five years who had full covariate information were used in the analysis. Spatial binomial logistic regression methodology was undertaken via Bayesian estimation based on Markov chain Monte Carlo (MCMC) using WinBUGS software version 1.4. The findings in this study revealed that 77% of children under five years in Guinea had anemia, and the prevalences in the regions ranged from 70.32% (Conakry) to 83.60% (NZerekore) across the country. After adjusting for non-spatial and spatial random effects in the model, older children (48–59 months) (OR: 0.47, CI [0.29 0.70]) were less likely to be anemic compared to those who are younger (0–11 months). Children whose mothers had completed secondary school or above had a 33% reduced risk of anemia (OR: 0.67, CI [0.49 0.90]), and children from household heads from the Kissi ethnic group are less likely to have anemia than their counterparts whose leaders are from Soussou (OR: 0.48, CI [0.23 0.92]).

## 1. Introduction

Anemia is a global public health problem affecting both developing and developed countries with major consequences for human health as well as social and economic development [1]. Most seriously affected are young children and women [2]. In 2011, the global prevalence of anemia in the world was 43% among children less than five years of age, and 67% in sub-Saharan Africa [3]. Anemia contributes to increased morbidity and mortality, reduced work productivity, and impaired neurological development [4,5,6]. According to Brabin et al. [7], the highest estimates of deaths attributed to anemia are for India and then sub-Saharan Africa. Health specialists note that the level of economic development has an influence on the level of endemicity [8]. Guinea is a West African country with a population of over 12 million inhabitants [9] and is not on the fringes of the impacts of anemia. Despite the efforts made by the Guinean Government and development partners to reduce prevalence among children, the rate is quite high. According to the Guinea Demographic and Health Survey (DHS) report in 2018, the prevalence rate was 75% among children under five years, and it varies widely from one geographical location to another, from 69% (Boke) to 78% (Faranah). Since 2012, the prevalence of anemia in children aged 6 to 59 months did not vary significantly, from 77% to 75% in 2018 [10].

In the medical manifest, it is pointed out that anemia in children is when the hemoglobin level in the blood is lower than 110 g/L due to iron deficiency [11]. It is characterized by signs which include pallor, abnormal tiredness, or repeated infections. Anemia during childhood has short and long-term effects on health. The former include an increased risk of morbidity due to infectious diseases [12,13]. In addition, anemia in children seriously affect the growth and mental development [14]. Thus, the consequences of anemia in children are multiple and harmful. They are reflected in human development (absences from class, school grade, repetitions, and poorer school performance), morbidity (bone disease, heart murmur, liver, and spleen enlargement), and mortality (significant death) [15,16].

In the context of health policy, mapping of disease incidence and prevalence is very important. Mainly, its aim is to smooth and predict certain health outcomes over a geographic domain of interest.

Spatial data analysis has a role to play in supporting the search for scientific explanation. It also has a role to play in more general problem solving because observations in geographic space are more than often correlated [17]. Within the framework of the development of decisions on the allocation of public health resources, the analysis helps decision makers in setting priorities.

A number of articles have demonstrated the various risk factors associated with anemia among children. For example, a multivariable hierarchical Bayesian geoadditive model which included a spatial effect for district of child’s residence was applied to examine the association of demographic, socio-economic, and environmental factors in four sub-Saharan African countries [18]. Another study was done on spatial pattern and determinants of anemia in Ethiopia. In that study, multilevel analysis was used, and spatial dependence is tested using Moran’s I statistic [19].

Despite the studies that have been carried out, there is limited literature on anemia in Guinea. Therefore, it is important to understand the risk factors of anemia in such a country. Thus, although overall the models are structurally similar, country specific applications help to understand the spatial distribution of a disease in much more detail with results directly applicable to health policy formulation.

## 2. Data and Methods

### 2.1. Study Data

The Guinea Multiple Indicator Cluster Survey (MICS5) was carried out in 2016 by National Institute of Statistics in collaboration with the National Malaria Control Program and the National Institute of Public Health, as part of round five of the MICS global survey program. Technical support was provided by the United Nations International Children’s Emergency Fund (UNICEF) and International Coach Federation (ICF) for testing for malaria and anemia in children under five years. Furthermore, UNICEF, United States Agency for International Development (USAID), the Global Fund/Catholic Relief Services (CRS), United Nations Population Fund (UNFPA), and United Nations Development Programme (UNDP) provided financial support to the project alongside the government. The objective of the survey was to provide valuable information for monitoring the progress made in Guinea for their international commitments.

The data were collected on 8081 households in the eight (8) regions of the country, including the capital, Conakry (Figure A1), with a response rate of 99%. The survey targeted men and women aged 15 to 49 years and children. Four sets of questionnaires were used in the survey, based on the standard MICS round 5 questionnaires developed by UNICEF and the standard questionnaires for Malaria Indicator Surveys (MIS) developed by ICF Macro within the framework of the international DHS program. These standard questionnaires were adapted to Guinean context. The types of questionnaires are: a household questionnaire which was used to collect demographic information on all members of the de jure household, the household, and the dwelling, an individual woman questionnaire administered in each household to all women aged 15 to 49 years, an individual questionnaire for children under five years administered to mothers for children under five years living in the household. It is in this questionnaire that the biomarker module (anemia and malaria test) was administered, and a verbal autopsy questionnaire administered to mothers for all children under five years who died during the past three years. The household, child, and woman questionnaires were used to extract information on anemia in children under five years, the characteristics of the household, and those of the mothers of the children [20].

### 2.2. Variables of Interest

#### 2.2.1. Response Variable

The dependent variable of the study is the status of anemia in children under five years. In the study, this disease was detected by screening, via an anemia test. The procedure consisted of collecting a drop of blood of the child in a microcuvette and then introducing it into the HemoCue photometer that showed hemoglobin level. All results are recorded in the child questionnaire in the section for biomarker testing. The response variable has two status: the positive status (the child has anemia) and the negative status (the child does not have anemia).

#### 2.2.2. Independent Variables

The independent variables used for the analysis are contextual, socio-economic, and demographic variables.

The contextual covariates were: natural region, administrative region and place of residence. The socio-economic and demographic variables: sex of the child, age of the child, sex of the household head, mosquito net observed in the house, status of mosquito net, mother’s education level, wealth index, ethnicity of the household head, religion of the household head, potable water source, household size, treatment of drinking water, access to electricity, own radio, own TV, main material of roof, main material of floor, wall exterior main material, and type of toilet.

### 2.3. Statistical Analysis

This present study aims to contribute to improving knowledge of anemia in children under five in Guinea in order to help the Government and its partners to better reorient and strengthen control strategies. To this end, the following approaches of analysis are used: description of the study population (frequencies and percentages), bivariate analysis (unadjusted), binary logistic regression modeling, and spatial analysis (models after adjusting for non-spatial random, spatial random effects) to describe heterogeneity of anemia in children in the space. Bayesian methodology, using Markov chain Monte Carlo (MCMC) methods, was used for parameters estimation in the models.

### 2.4. Statistical Modeling

#### 2.4.1. Models Specification

Let $Y_{ij}$ be disease status of child *i* in region *j*, j=1,2,…,8, and $i=1,2,\ldots ,n_{j}$, where $n_{j}$ is the number of children in region *j*. We have binary responses, such as Equation (Equation 1),
(1)$$Y_{ij}=\left\{\begin{array}{c} 1,\;\mathrm{if}\;\mathrm{child}\;i\;\mathrm{in}\;\mathrm{region}\;j\;\mathrm{is}\;\mathrm{anemia}\;\mathrm{positive}\\ 0,\;\mathrm{otherwise}. \end{array}\right.$$

This study assumes that the dependent variable $Y_{ij}$ is Bernoulli distributed, i.e., $Y_{ij}$∣$p_{ij}$∼ Bernoulli $\left(p_{ij}\right)$ with an unknown mean $E\left(Y_{ij}\right)=p_{ij},$ being related to the independent variables as follows:

Equation (Equation 2) (M1): Logistic regression model, a direct linkage exists between a linear predictor and the parameter of interest.
(2)$$\begin{array}{cc} g\left(E\left(Y_{ij}\right)\right) & =\log\left(\frac{P\left(Y_{ij}=1|x\right)}{1-P\left(Y_{ij}=1|x\right)}\right)\\ \; & =X^{T}\beta \end{array}$$

Equation (Equation 3) (M2): Non-spatial random effects (region-specific) $v_{j}$ is include in the model.
(3)$$\begin{array}{cc} g\left(E\left(Y_{ij}\right)\right) & =\log\left(\frac{P\left(Y_{ij}=1|x,v_{j}\right)}{1-P\left(Y_{ij}=1|x,v_{j}\right)}\right)\\ \; & =X^{T}\beta +v_{j} \end{array}$$

Equation (Equation 4) (M3): Spatial random effects $u_{j}$ is include in the model.
(4)$$\begin{array}{cc} g\left(E\left(Y_{ij}\right)\right) & =\log\left(\frac{P\left(Y_{ij}=1|x,u_{j}\right)}{1-P\left(Y_{ij}=1|x,u_{j}\right)}\right)\\ \; & =X^{T}\beta +u_{j} \end{array}$$

Equation (Equation 5) (M4): Convolution model, both non-spatial and spatial random effects are include in the model.
(5)$$\begin{array}{cc} g\left(E\left(Y_{ij}\right)\right) & =\log\left(\frac{P\left(Y_{ij}=1|x,u_{j},v_{j}\right)}{1-P\left(Y_{ij}=1|x,u_{j},v_{j}\right)}\right)\\ \; & =X^{T}\beta +u_{j}+v_{j},\;i=1,\ldots ,n_{j},\\ and\;j=1,\ldots ,8. \end{array}$$

In Equations (Equation 2)–(Equation 5), $X^{T}$ is a k-dimensional row-vector of covariates with β as the corresponding vector of regression coefficients.

Spatial random effects *u*, which determines the spatial auto-correlation, occurs when adjacent regions are more related to each other than more distant regions [21]. This effect may occur in social research, since the surrounding areas have similar social, economic, and cultural characteristics [22]. In order to provide information about the spatial structure of the data, detecting spatial dependence may be useful. Regarding the non-spatial component, random effects *v* is included to account for the heterogeneity, over-dispersion, and arbitrary choice of a spatial unit. Such an effect is related to the spatial differentiation of geographic units. In these situations, the variability under the assumed distribution may be greater than expected [23]. This non-spatial random effects would correct and smooth the distribution if they are included in model [24].

The vector *v* follows a prior normal distribution with a vector of mean 0, and a variance-covariance matrix $\sigma^{2}I$ (with I being identity matrix and $\sigma^{2}>0$ unknown). Concerning the spatial component *u*, we assume that the prior is represented by a Markov Gaussian field or conditional Gaussian autoregressive model [25]. In this case, let $u_{-j}$ denote the vector of effects excluding that of the *j*-th region, then we assume Equation (Equation 6) as follows,
(6)$$u_{j}|u_{-j}∼N\left(\frac{1}{n_{j}}\sum_{r∼j}u_{r},\frac{\tau_{u}^{2}}{n_{j}}\right)$$
where $n_{j}$ is the number of neighborhoods of region *j*, the expression r∼j denotes all units *r* neighborhoods of area *j*, and $\tau_{u}$ is the standard deviation parameter. We assume inverse gamma hyperpriors for the variance of the normal priors.

#### 2.4.2. Parameters Estimation

Bayesian MCMC simulation entails estimating the posterior distribution of all parameters by combining prior information on them with the likelihood for the respective model, and sampling each parameter sequentially from its conditional distribution.

The posterior distribution is factored as Equation (Equation 7) below,
(7)$$\begin{array}{cc} \pi (u,v,\tau^{2},\sigma^{2},\beta |y_{1},\cdots ,y_{n_{j}}) & \propto \left[\prod_{j=1}^{8}\prod_{i=1}^{n_{j}}L\left(Y_{ij}|u,v,\tau^{2},\sigma^{2},\beta \right)\right]\times \left[\prod_{j=1}^{8}\pi \left(v_{j}|\sigma^{2}\right)\pi \left(\sigma^{2}\right)\right]\\ \; & \times \left[\prod_{j=1}^{8}\pi \left(u_{j}|u_{-j},\tau^{2}\right)\pi \left(\tau^{2}\right)\right]\times \left[\prod_{h=1}^{k}\pi \left(\beta_{h}\right)\right] \end{array}$$

The conditional distributions are generically denoted by π(o∣y), and the contribution to likelihood of the *i*-th unit in the *j*-th area by $L_{ij}\left(Y_{ij}\right),i=1,\ldots n_{j}$ (where $n_{j}$ represents the number of observations in the *j*-th region). Prior distributions for fixed and random effects and hyperpriors are mutually independent. Furthermore, conditional on explanatory variables and on the set of parameters, observations are independent.

In Bayesian approach, the prior distribution expresses past knowledge of the parameters, or the complete ignorance of such past knowledge in the situation of prior distributions with high variability. In this study, we used non-informative priors for the intercept and the coefficients (normal prior with mean = 0 and precision, the inverse of variance = 1 $\times 10^{-3}$). For both non-spatial and spatial random effects, also non-informative, were imposed on their inverse variance (gamma distributions with delimiting values = 1 $\times 10^{-3}$ and 1 $\times 10^{-3}$).

Parameters estimation of the models was done by Markov chain Monte Carlo (MCMC) simulation technique. The number of MCMC chains was 100,000 iterations with a burn in period of 5000 iterations.

In order to check the convergence of the simulated sequences in the models, we used the convergence diagnostic $\hat{R}$ of Gelman and Rubin [23], which was close to 1 for all parameters. Furthermore, the trace plots of these parameters show the convergence of the Markov chains (Figure A2).

#### 2.4.3. Diagnostics of Model

The Deviance Information Criterion (DIC) was used to compare models as suggested by Spiegelhalter et al. [26]. DIC value is given by this Equation (Equation 8),
(8)$$\begin{array}{c} \mathrm{DIC}=\bar{D}+\mathrm{pD}=\hat{D}+2\mathrm{pD}. \end{array}$$
$\bar{D}$ is the posterior mean of the deviance, which is a measure of goodness of fit statistic for a statistical model and pD = $\bar{D}-\hat{D}$ is the effective number of parameters.

The model with the smallest DIC is the best fitting model. According to Spiegelhalter et al. [26,27], by comparing the models, a difference in DIC of 3 or less between two models cannot be distinguished, while for a difference of between 3 and 7, the two models can be weakly differentiated.

## 3. Results

### 3.1. Description of the Study Population

In total, 2609 children under five years who had full covariate information were used in the current analysis.

Table 1 shows that 77% of the children are positive for anemia and about 15% for malaria.

Analysis by sex showed that there were more male children (51.36%) and more male headed households (85.17%) than female. The distribution of children by place of residence shows that 71.18% are in rural areas and only 28.82% live in urban areas. However, children who are in big cities and those in secondary cities are almost equal, (14.64%) and (14.18%), respectively. Regarding the analysis by administrative region, the majority of children are in the region of Boke (17.90%), followed by Nzerekore (16.67%), and the region of Mamou has the lowest percentage (8.78 %). In terms of natural region, 24.76 % of children lived in Maritime Guinea, 22.35% in Middle Guinea, 22.08% in Upper Guinea, 20.54% in Forested Guinea, and 10.27% in Conakry. The highest percentage of children (29.86%) were between 48 and 59 months, and the lowest (6.52%) were between 0 and 11 months. According to the socio-economic status of the household (Wealth index), 24.68% had a disadvantaged standard of living (poor level) and 12.04% of households had a good level (rich level). Additionally, 74.74% of the mothers of the children and 66.73% of household heads have not attended formal education. In the majority of the households (98.93%), mosquito nets were hung up, and 98.43% of the nets observed were in good conditions. The ethnic group distribution of the household heads shows that 36.80% were Peul, 25.49% were Malinke, 16.67% were Soussou, 7.17% were Guerze or Kono or Mano, 5.98% were Kissi, 2.38% were Toma, and 5.52% other ethnicity. It is the household heads of the Muslim religion who are more represented (84.44%). In terms of potable water source in the households, 78.42% of them had improved water sources. Thus, 66.73% did not treat drinking water. The analysis of the number of people in the household shows that households with 1–5 people are 40.17%, 36.14% are between 6 and 8, and 23.69% have 9 or more people. 27.06% of households have access to electricity. The household heads who had their own radio and television were 48.41% and 25.53%, respectively. The main materials most used as roof, wall exterior, and floor were metal sheets (74.01%), cement, stone with lime cement, brick, or cement block (70.18%), and cement, grout, or carpet (51.59%), respectively.

Table 2 shows the findings of the bivariate analysis (anemia in children versus administrative region) and binary logistic model by including non-spatial random effects. These results indicate that the prevalence of anemia among children varies according to the region of residence. From the bivariate analysis, the region of Nzerekore (85.29%) has the highest prevalence. On the other hand, in the region of Labe, the prevalence is lower (69.39%).

Table 3 presents the results of bivariate analysis (unadjusted) and binary logistic regression. All the interpretations of the models were done using the odds ratio and corresponding 95% credible intervals.

In view of the results of bivariate analysis, we noticed that the variables associated with the status of anemia in children are: place of residence, administrative region, natural region, age of the child, standard of living of the household, mother’s level of education, ethnicity of household head, religion of household head, household’s access to electricity, and whether the household head has their own television.

Indeed, children from rural areas were more likely to be anemic (OR: 1.59, CI [1.31 1.93]) when compared to those from urban areas. The same observation was made by comparing children from rural areas and those from big cities. Rural children were more likely to have anemia (OR: 1.59, CI [1.24 2.03]). The results also indicate that children in the region of Nzerekore were more likely to have anemia (OR: 1.54, CI [1.09 2.18]) compared to those from Boke region. It appears that children from Conakry were less likely to be anemic (OR: 0.61, CI [0.44 0.84]) than those of Maritime Guinea. In addition, children in the age group of 48–59 months were less likely to be anemic (OR: 0.51, CI [0.34 0.78]) than children in the 0–11 months age group. Regarding education level of mother, the analysis shows that the children of mothers in an advanced level (secondary school or above) were less likely to have anemia (OR: 0.61, CI [0.47 0.79]) compared to the children of mothers that do not have formal education (no educational attainment). As for the standard of living of the household, children from rich households were less likely to be anemic (OR: 0.52, CI [0.38 0.71]) compared to their counterparts in poor households. Children whose household head is of the Peul ethnic group were less likely to have anemia (OR: 0.66, CI [0.50 0.86]) compared to the children whose head is Soussou. Results also indicate that children whose household head is animist or no religion were less likely to be anemic (OR: 0.80, CI [0.61 1.04]) compared to the children whose head is Muslim. If the household did not have electricity, children in the household were more likely to have anemia (OR: 1.50, CI [1.23 1.83]) than those in the household with electricity. Children whose households do not own a television are more likely to have anemia (OR: 1.51, CI [1.23 1.84]) compared to those whose households own a television.

### 3.2. Model without Spatial and Non-Spatial Components

Only variables that were significantly associated with anemia in children were included in the binary logistic model (model M1). The results also confirm that the variables administrative region, age of the child, education level of mother, and the ethnicity of the household head are significantly associated with the status of anemia among children. Therefore, children in the region of Labe were less likely to have anemia (OR: 0.65, CI [0.44 0.96]) than their counterparts in the region of Boke. According to the child’s age, children in the age group 48–59 months were less likely to be anemic (OR: 0.46, CI [0.30 0.71]) than those in the 0–11 months group. Furthermore, children of mothers with a secondary school qualification or above were less likely to be anemic (OR: 0.67, CI [0.50 0.91]) than those of mothers with no educational attainment. Finally, comparing the level of risk of children according to the ethnicity of their household head shows that children who are under the responsibility of a household head from the Kissi ethnic group were less likely to be anemic (OR: 0.45, CI [0.22 0.92]) than those under the responsibility of the Soussou ethnic group.

### 3.3. Model Assessment and Comparison

Four models are provided, which are M1, M2, M3, and M4. M1 is the binary logistic regression, M2 is the model after adjusting for non-spatial random effects (region- specific), M3 is the model after adjusting for spatial random effects, and M4 is the convolution model (with both non-spatial and spatial random effects). The DIC values were used to compare the goodness of fit of these four separately models M1, M2, M3, and M4 in explaining variations of children anemia. Model with a small DIC value provides a better fit. By comparing their DICs, models two (M2) and four (M4) are the preferred models. They have the same and the smaller DIC (2721.9). Indeed, extension of model M1, to Model M2 by including non-spatial random effects and model M4 by including both non-spatial and spatial random effects improved the goodness of fit of the final model. Note that the three models (M2, M3, and M4) are not significantly different form each other as the difference in DIC is less than 3 [26,27]. So the three models have the same factors associated with anemia among children under five years across the country.

### 3.4. Factors Associated with Anemia in Children from the Spatial Models

In Table 4, we have the factors associated with anemia among children in Guinea after controlling for the non-spatial random effects (M2), spatial random effects (M3), and both non-spatial and spatial random effects (M4). These models were implemented using WinBUGS version 1.4 (MRC Biostatistics Unit, Cambridge, UK)

Significant risk factors shown in Figure 1 of the model incorporating the non-spatial random effects (M2) were included in the binary logistic model. For example, in the model (M1), children aged 48–59 months were less likely to have anemia (OR: 0.46, CI [0.30 0.71]) compared to those who are younger (0–11 months). Adjustment for non-spatial random effects have provided a protective effect against anemia for this age group of children, which reduces the odds of being anemic by 53% (OR: 0.47, CI [0.29 0.70]). Children whose mothers attained secondary school or above in education had a reduced chance of being anemia positive 33% (OR: 0.67, CI [0.49 0.90]) compared to children of mothers who did not have formal education (no educational attainment). Children who are under the responsibility of household heads from Peul ethnic group after controlling for non-spatial random effect were associated with anemia among children as well. They are less likely to have anemia (OR: 0.57, CI [0.41 0.78]) than their counterparts whose leaders are Soussou.

In view of the importance of the mother’s education level and the ethnicity of household head factors, the stratified results for these variables are shown by the standard of living of the household in Figure A3.

Figure 2 illustrates the results of the model by including non-spatial random effects (M2). We have areas that are perceived as high and low prevalence of anemia among children. The Nzerekore region (yellow color) has the highest prevalence, and the Conakry region (maroon color) has the lowest prevalence. However, seeing these prevalences, all areas are considered to have a high prevalence of anemia among children. This map was obtained from the results of the Bayesian analysis.

Appendix ATable A1 presents the posterior means (relative risk) and standard deviation (sd) for non-spatial random *v* and spatial random effects *u*. The results presented as relative risk of the non-spatial random effects for the best fitting model are given in the map (Figure A4). The map shows that the region of Nzerekore (yellow color) has high relative risk (0.11). Moreover, a few clusters (Conakry, Boke, and Kindia ) with moderate relative risk (0.0 to 0.1) are seen in the map (green color). Regarding the posterior standard deviations of the non-spatial random effects, the results show that the region of Nzerekore tends to be higher (0.14) than the other regions. This means the within-region variation of anemia tends to be higher than the rest of the regions after accounting for all the covariate effects. The map of the posterior means for spatial random effects also shows that the regions of Nzerekore, Boke, Kindia, and Conakry (green color) had a moderate relative risk, between 0.0 and 0.1 (Figure A4).

## 4. Discussion

Anemia is becoming a major cause of rising child mortality under five years. The findings in this study could be relevant information for control programs aimed at reducing the prevalence of anemia in Guinea, especially in the regions with high prevalences. Improving children’s nutritional status would help to prevent child deaths. As a result, it would reduce the rate of child mortality while also helping in the achievement of the Sustainable Development Goals (SDG 3).

However, the study contributes to the literature, and the models can be replicated in other countries with additional factors that the study did not account for.

The results suggest that the association between the status of anemia among children and some variables (place of residence, own TV, and access to electricity) was significant in the bivariate analysis, but non-significant in the models M1, M2, M3, and M4. Furthermore, we note that only the region of Labe was significantly associated with anemia in the binary logistic model (M1). Efforts to control anemia among children under five years should focus on factors such as the age of the child, the mother’s level of education, and the ethnicity of the household head. The results obtained are consistent not only within country context but what has been shown in previous studies. The authors have showed that the child’s late age and his mother’s high-level education were negatively associated to childhood anemia [28,29]. The variation in the child’s age determines the hemoglobin requirements of red blood cells for physical and psychomotor functioning, as well as cognitive development in children in their early years of life [30]. Children whose mothers are educated are less likely to be ill. In general, more educated mothers are more likely to take children to hospitals [31]. In addition, Flores et al. [32] have attributed the ethnicity of the parents as an important factor on children diseases. The very different lifestyles of the various ethnic groups and the geographical setting in which an ethnic group resides could strongly influence the health of the child.

The overall prevalence of malaria infection in children under five years of age was estimated at 15.14%. According to studies, it is becoming increasingly evident that human diseases are not isolated from one another [33]. Among many other factors, malaria plays a major causative role of anemia globally. The mechanisms causing anemia during malaria are extremely diverse, involving immunological factors that act differently depending on age and malaria epidemiology [34]. Almost all infants and young children have a reduced hemoglobin level in areas where malaria is prevalent [35]. In those affected by malaria, the blood is infected and the result is an abnormal drop in the number of red blood cells, which compromises the rapid recovery from anemia.

In our study, a Bayesian spatial modeling was applied that allowed for understanding of disease factors and variations in different regions. This framework could be used to investigate a variety of nutritional or pediatric diseases. The utility of such a method is to express prior knowledge about population parameters in order to guide the statistical inference process, or to express complete ignorance of such prior knowledge. This approach also implies that the estimate borrows strength from both neighboring observation data and auxiliary data on neighborhood characteristics.

The findings of the Bayesian analysis framework confirmed a relationship between the child’s age, the mother’s level of education, the ethnicity of the household head, and an increased risk of anemia in children. The incorporation of random effects helped to avoid underestimating the standard errors of model parameters, thereby avoiding wrong statistical significance of covariates since their credible intervals would be deceptively narrower [36]. For example, in the case of ethnicity of household head factor, Kissi ethnic category coefficient had lower disease odds, 55% (OR:0.45, CI [0.22 0.92] in model M1, where non-spatial random effects were not incorporated. However, this coefficient was increased (OR:0.48, CI [0.22 0.91] in Models M2, M3, and M4 when the non-spatial random, spatial random effects, and both, respectively, were included in the models, the chance of having anemia is reduced by approximately 52%. In addition, the analysis showed a heterogeneity of the spatial distribution of anemia among children. The region of Nzerekore is identified as a high prevalence region, which should raise concerns to policy-makers.

This study has some limitations, as anemia in children could also take into account the association with biological factors, food security, and different foods eaten in the household. For example, anemia among children is frequently associated with many aspects such as mothers received iron supplementation during pregnancy [37]. However, the current study used only demographic and socio-economic factors. Then, other socio-economic factors were not included in the models like economic activity, social class, and income of mother. The relationship was also statistical association, not causal between investigated factors and anemia. MICS data is a cross-sectional study, this could neither establish temporality nor causality of the observed associations with the anemia of children. Moreover, the results could be influenced by sample error. In terms of methodology, the Bayesian computational approach we adopted enables us to provide estimates of parameters in an otherwise too complex model. It also allows us to refrain from the assumption of mutual independence between areas usually imposed in multilevel statistical models [38].

## 5. Conclusions

In conclusion, this study applied a Bayesian methodology using MCMC methods. The objective was to identify the factors associated with anemia and to map their possible spatial effects on anemia among children under five years of age. The analysis revealed that children in Guinea with a lower age are at higher risk of anemia. Child’s age effect suggests the importance of paying attention to child feeding practices, especially when the child is very young. The analysis also showed that children’s anemia is influenced by the level of education of mothers and the ethnicity of the household head. The children of mothers with a higher education level were more protected against anemia. Children under the responsibility of Kissi and Peul’s household heads were less likely to have anemia than their counterparts whose leader is Soussou. The findings from spatial analysis also highlighted that Nzerekore region had the higher prevalence of anemia among children. More emphasis should be placed on mother education as well as community sensibilization in areas where children are more affected. The results may help policy makers to identify regions that require more attention to reduce prevalence of anemia in Guinea.

## Figures and Tables

**Figure 1 ijerph-18-06447-f001:**
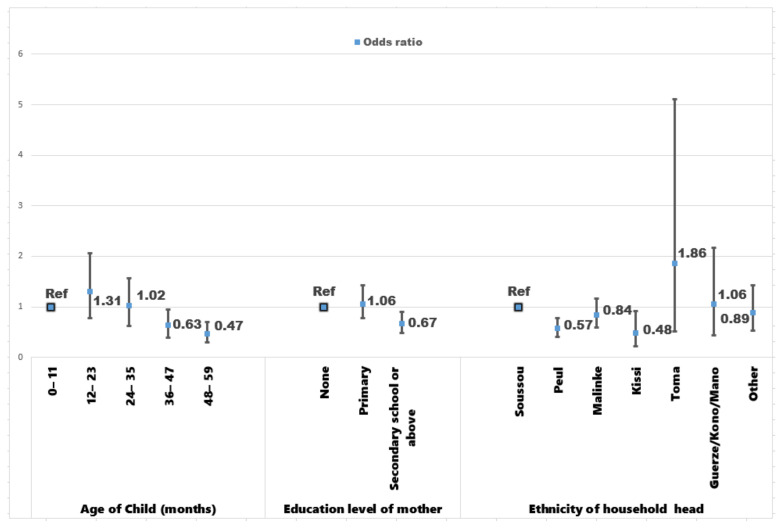
Factors associated with anemia from the model by including non-spatial random effects.

**Figure 2 ijerph-18-06447-f002:**
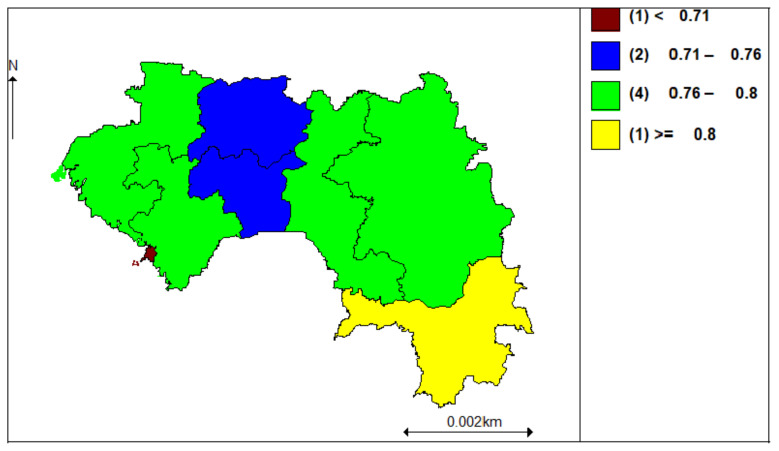
Prevalence categories of anemia.

**Table 1 ijerph-18-06447-t001:** Socio-economic and demographic characteristics of the respondents.

Variables	N	Percentage (%)
Status of anemia		
Negative	600	23.00
Positive	2009	77.00
Status of malaria		
Negative	2214	84.86
Positive	395	15.14
Sex of child		
Male	1340	51.36
Female	1269	48.64
Place of residence		
Big city	382	14.64
Secondary city	370	14.18
Rural	1857	71.18
Place of residence		
Urban	752	28.82
Rural	1857	71.18
Administrative Region		
Boke	467	17.90
Conakry	268	10.27
Faranah	369	14.14
Kankan	308	11.81
Kindia	288	11.04
Labe	245	9.39
Mamou	229	8.78
NZerekore	435	16.67
Natural Region		
Maritime Guinea	646	24.76
Middle Guinea	583	22.35
Upper Guinea	576	22.08
Forested Guinea	536	20.54
Conakry	268	10.27
Age of Child (months)		
0–11	170	6.52
12–23	427	16.37
24–35	523	20.05
36–47	710	27.21
48–59	779	29.86
Sex of the household head		
Male	2222	85.17
Female	387	14.83
Education level of household head		
None	1741	66.73
Primary	288	11.04
Secondary school or above	580	22.23
Mosquito net observed in the house		
Observed hanging	2581	98.93
Observed not hanging	28	1.07
Status of mosquito net		
Good	2568	98.43
Bad	41	1.57
Education level of mother		
None	1950	74.74
Primary	329	12.61
Secondary school or above	330	12.65
Ethnicity of household head		
Soussou	435	16.67
Peul	960	36.80
Malinke	665	25.49
Kissi	156	5.98
Toma	62	2.38
Guerze/Kono/Mano	187	7.17
Other	144	5.52
Religion of household head		
Muslim	2203	84.44
Christian	353	13.53
Others (Animist, no religion)	53	2.03
Wealth index		
Poorest	644	24.68
Second	671	25.72
Middle	522	20.01
Fourth	458	17.55
Richest	314	12.04
Potable water source		
improved water source	2046	78.42
Non-improved water source	563	21.58
Total member in the house		
1–5	1048	40.17
6–8	943	36.14
9 and more	618	23.69
Treatment of drinking water		
Yes	868	33.27
No	1741	66.73
Access to electricity		
Yes	706	27.06
No	1903	72.94
Own Radio		
Yes	1263	48.41
No	1346	51.59
Own TV		
Yes	666	25.53
No	1943	74.47
Main material of roof		
Palm leaf/Palm/Bamboo/wood/wooden planks/Cardboard	239	9.16
Grass	367	14.07
Metal sheet	1931	74.01
Other (Roof tiles/Concrete/Cement/Mat/ shingles, no roof)	72	2.76
Main material of floor		
Earth/Sand/Cow dung	998	38.25
Plank of wood/bamboo (Plank of wood, palm/bamboo/Other)	56	2.15
Floor tile (floor tile/ waxed wood/vinyl/asphalt)	209	8.01
Cement/grout/Carpet	1346	51.59
Wall exterior main material		
Clods of earth	547	20.97
Bamboo with Mud	100	3.83
Stone with mud	83	3.18
Cement/Stone with lime cement/Brick/Cement block	1831	70.18
Wood(stick/trunk/plywood/cardboard/salvage wood/wood planks)	11	0.42
Other (no wall/adobe/ covered/Adobe not covered)	37	1.42
Type of toilet		
Improved toilet	1687	64.66
Non-improved toilet	922	35.34

**Table 2 ijerph-18-06447-t002:** Prevalence of anemia by region.

	Prevalence N (%)	Predicted Prevalence (after Adjusting for Non-Spatial Random Effects) (%)
Total	2609 (77.00)	-
Boke	467 (79.01)	77.98
Conakry	268 (70.52)	70.32
Faranah	369 (77.24)	77.71
Kankan	308 (75.65)	77.30
Kindia	288 (80.56)	78.96
Labe	245 (69.39)	72.23
Mamou	229 (69.87)	71.40
Nzerekore	435 (85.29)	83.60

**Table 3 ijerph-18-06447-t003:** Factors associated with anemia in the eight (8) regions in Guinea.

	Crude Odds Ratio (95% CI)	OR (95% CI), M1
Place of residence		
Urban	1	1
Rural	1.59 [1.31 1.93]	1.35 [0.96 1.90]
Place of residence		
Big city	1	-
Secondary city	0.99 [0.73 1.36]	-
Rural	1.59 [1.24 2.03]	-
Administrative Region		
Boke	1	1
Conakry	0.64 [0.45 0.90]	0.94 [0.62 1.42]
Faranah	0.90 [0.65 1.25]	0.82 [0.56 1.20]
Kankan	0.83 [0.59 1.16]	0.73 [0.47 1.13]
Kindia	1.10 [0.76 1.59]	1.05 [0.71 1.56]
Labe	0.60 [0.42 0.86]	0.65 [0.44 0.96]
Mamou	0.62 [0.43 0.88]	0.73 [0.49 1.08]
Nzerekore	1.54 [1.09 2.18]	1.25 [0.77 2.04]
Natural Region		
Maritime Guinea	1	-
Middle Guinea	0.63 [0.49 0.82]	-
Upper Guinea	0.87 [0.66 1.15]	-
Forested Guinea	1.21 [0.90 1.63]	-
Conakry	0.61 [0.44 0.84]	-
Age of Child (months)		
0–11	1	1
12–23	1.26 [0.79 2.03]	1.26 [0.78 2.04]
24–35	1.03 [0.66 1.62]	0.99 [0.63 1.57]
36–47	0.68 [0.44 1.03]	0.62 [0.40 0.95]
48–59	0.51 [0.34 0.78]	0.46 [0.30 0.71]
Wealth index		
Poorest	1	1
Second	1.02 [0.77 1.34]	0.88 [0.66 1.17]
Middle	0.75 [0.57 0.99]	0.75 [0.55 1.00]
Fourth	0.67 [0.50 0.89]	0.68 [0.41 1.14]
Richest	0.52 [0.38 0.71]	0.57 [0.30 1.11]
Education level of mother		
None	1	1
Primary	1.08 [0.81 1.44]	1.05 [0.77 1.42]
Secondary school or above	0.61 [0.47 0.79]	0.67 [0.50 0.91]
Ethnicity of household head		
Soussou	1	1
Peul	0.66 [0.50 0.86]	0.65 [0.47 0.90]
Malinke	0.84 [0.62 1.13]	0.90 [0.61 1.34]
Kissi	0.97 [0.61 1.53]	0.45 [0.22 0.92]
Toma	2.33 [0.97 5.59]	1.35 [0.44 4.13]
Guerze/Kono/Mano	1.42 [0.89 2.26]	0.89 [0.39 2.01]
Other	0.99 [0.61 1.59]	0.88 [0.54 1.46]
Religion of household head		
Muslim	1	1
Christian	0.82 [0.63 1.06]	1.60 [0.85 3.04]
Others (Animist, no religion)	0.80 [0.61 1.04]	0.72 [0.27 1.97]
Own TV		
Yes	1	1
No	1.51 [1.23 1.84]	1.05 [0.70 1.59]
Access to electricity		
Yes	1	1
No	1.50 [1.23 1.83]	0.88 [0.58 1.34]
Deviance information criterion (DIC)		2724.9

**Table 4 ijerph-18-06447-t004:** Factors associated with anemia from the models after adjusting the non-spatial, spatial random effects and convolution model.

	OR (95% CI), M2	OR (95% CI), M3	OR (95% CI), M4
Place of residence			
Urban	1	1	1
Rural	1.31 [0.92 1.82]	1.31 [0.93 1.80]	1.32 [0.93 1.82]
Age of Child (months)		
0–11	1	1	1
12–23	1.31 [0.78 2.06]	1.32 [0.78 2.06]	1.30 [0.77 2.04]
24–35	1.02 [0.62 1.57]	1.03 [0.62 1.57]	1.02 [0.62 1.56]
36–47	0.63 [0.39 0.95]	0.63 [0.39 0.95]	0.63 [0.39 0.93]
48–59	0.47 [0.29 0.70]	0.47 [0.30 0.70]	0.47 [0.29 0.70]
Wealth index			
Poorest	1	1	1
Second	0.89 [0.66 1.17]	0.89 [0.66 1.17]	0.89 [0.66 1.17]
Middle	0.75 [0.55 1.01]	0.75 [0.55 1.00]	0.75 [0.55 1.00]
Fourth	0.72 [0.42 1.16]	0.72 [0.42 1.15]	0.71 [0.42 1.15]
Richest	0.63 [0.31 1.13]	0.62 [0.31 1.11]	0.62 [0.31 1.12]
Education level of mother			
None	1	1	1
Primary	1.06 [0.78 1.43]	1.06 [0.78 1.43]	1.06 [0.78 1.43]
Secondary school or above	0.67 [0.49 0.90]	0.67 [0.49 0.90]	0.67 [0.49 0.90]
Ethnicity of household head			
Soussou	1	1	1
Peul	0.57 [0.41 0.78]	0.56 [0.41 0.76]	0.58 [0.42 0.79]
Malinke	0.84 [0.59 1.17]	0.83 [0.58 1.17]	0.86 [0.59 1.21]
Kissi	0.48 [0.22 0.91]	0.48 [0.22 0.91]	0.48 [0.23 0.92]
Toma	1.86 [0.52 5.11]	1.87 [0.53 5.13]	1.84 [0.52 4.99]
Guerze/Kono/Mano	1.06 [0.44 2.17]	1.07 [0.44 2.19]	1.05 [0.44 2.16]
Other	0.89 [0.53 1.43]	0.88 [0.53 1.41]	0.89 [0.53 1.43]
Religion of household head			
Muslim	1	1	1
Christian	1.81 [0.90 3.30]	1.83 [0.91 3.33]	1.79 [0.90 3.24]
Others (Animist, no religion)	0.95 [0.31 2.32]	0.96 [0.31 2.34]	0.93 [0.30 2.28]
Own TV			
Yes	1	1	1
No	1.10 [0.71 1.62]	1.10 [0.71 1.62]	1.09 [0.71 1.61]
Access to electricity			
Yes	1	1	1
No	0.91 [0.59 1.34]	0.91 [0.59 1.33]	0.91 [0.59 1.35]
Deviance information criterion (DIC)	2721.9	2722.9	2721.9

## Data Availability

Publicly available datasets were analysed in this study. This data can be found here: https://mics.unicef.org/surveys (accessed on 27 May 2020).

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
