# Peer review of "Bayesian Spatial Modeling of Anemia among Children under 5 Years in Guinea"

_ijerph, 2021, doi:10.3390/ijerph18126447_

Round 1

Reviewer 1 Report

The manuscript presented here by Barry et al. is very detailed and used large cross-section of data and elaborate variables to understand the factors affecting children with Anemia. Interesting is the discrepancy of anemia in the different ethnic groups which could be either due to socioeconomic factors or food choices among the ethnic groups. I hope this interesting analysis could be a ground report for improving the situation in the country. 

In the discussion section, a more detailed account of how this framework could be useful to study different nutritional or other pediatric diseases could be elaborated by the authors.  

Author Response

Response to Reviewer 1 Comments and Suggestions

Point 1: The manuscript presented here by Barry et al. is very detailed and used large cross-section of data and elaborate variables to understand the factors affecting children with Anemia. Interesting is the discrepancy of anemia in the different ethnic groups which could be either due to socioeconomic factors or food choices among the ethnic groups. I hope this interesting analysis could be a ground report for improving the situation in the country. 

In the discussion section, a more detailed account of how this framework could be useful to study different nutritional or other pediatric diseases could be elaborated by the authors. 

Response 1:

In the discussion part, I added some details of how this framework could be useful to study different nutritional or other pediatric diseases.

Line 334. In our study, a Bayesian spatial modeling was applied that allowed for understanding of disease factors and variations in different regions. This framework could be used to investigate a variety of nutritional or pediatric diseases.  The utility of such a method is to express prior knowledge about population parameters in order to guide the statistical inference process, or to express complete ignorance of such prior knowledge. This approach also implies that the estimate borrows strength from both neighboring observation data and auxiliary data on neighborhood characteristics.

Line 304 Improving children's nutritional status would help to prevent child deaths.

As a result, it would reduce the rate of child mortality while also helping in the achievement of the Sustainable Development Goals (SDG 3).

Reviewer 2 Report

Journal: International Journal of Environmental Research and Public Health

Manuscript Number: ijerph-1166701

Title: Bayesian Spatial Modeling of Anemia among Children under 5 years in Guinea

The goal of this work was to identify the factors associated with anemia among children under 5 years of age in Guinea and to map their possible spatial effects. Spatial binomial logistic regression methodology was undertaken via Bayesian estimation based on Markov chain Monte Carlo (MCMC). The analysis revealed that children in Guinea with a lower age are at higher risk of anemia. The analysis also showed that children’s anemia is influenced by the level of education of mothers and the ethnicity of the household head. The children of mothers with a higher education level were more protected against anemia.

Data from the Guinea Multiple Indicator Cluster Survey (MICS5) 2016 was used for this study. A total of 2609 children under 5 years who had full covariate information were used in the analysis.

The design is solid, clear, the results are convincing. The geographical results are interesting.

MAJOR

In Abstract, they mentioned that “Anemia is a major public health problem in Africa with an increasing number of children under 5 years getting infected” and in Section 3.4: #...associated with anemia infection”

Anemia is not an infectious disease. Anemia is a decrease in the total amount of red blood cells (RBCs) or hemoglobin in the blood. Causes of decreased production include iron deficiency, vitamin B12 deficiency, folate deficiency, thalassemia, and a number of neoplasms of the bone marrow. Then anemia is problem of malnutrition. Causes of increased breakdown include genetic conditions such as sickle cell anemia, infections such as malaria (they measure nets against mosquitoes), and certain autoimmune diseases.

My suggestion is to clarify the definition of anemia.

Minor

The agreement between the estimated prevalence of anemia and the predicted by the models (Table 2) is outstanding. I would prefer to convert Table 2 into a Figure.

The labels of Figure 1 are hardly noticeable.

The blue color in the map of Figure 2 is indistinguishable.

Impossible to read the labels of Figure 2A.

Author Response

Response to Reviewer 2 Comments and Suggestions

The goal of this work was to identify the factors associated with anemia among children under 5 years of age in Guinea and to map their possible spatial effects. Spatial binomial logistic regression methodology was undertaken via Bayesian estimation based on Markov chain Monte Carlo (MCMC). The analysis revealed that children in Guinea with a lower age are at higher risk of anemia. The analysis also showed that children’s anemia is influenced by the level of education of mothers and the ethnicity of the household head. The children of mothers with a higher education level were more protected against anemia.

Data from the Guinea Multiple Indicator Cluster Survey (MICS5) 2016 was used for this study. A total of 2609 children under 5 years who had full covariate information were used in the analysis.

The design is solid, clear, the results are convincing. The geographical results are interesting.

Point 1: In Abstract, they mentioned that “Anemia is a major public health problem in Africa with an increasing number of children under 5 years getting infected” and in Section 3.4: #...associated with anemia infection”

Anemia is not an infectious disease. Anemia is a decrease in the total amount of red blood cells (RBCs) or hemoglobin in the blood. Causes of decreased production include iron deficiency, vitamin B12 deficiency, folate deficiency, thalassemia, and a number of neoplasms of the bone marrow. Then anemia is problem of malnutrition. Causes of increased breakdown include genetic conditions such as sickle cell anemia, infections such as malaria (they measure nets against mosquitoes), and certain autoimmune diseases.

My suggestion is to clarify the definition of anemia.

Response 1:  I agree that anemia is not an infectious disease

I rewrote the sentences as follows: 

Line 1 (Abstract).  Anemia is a major public health problem in Africa, affecting an increasing number of

 children under 5 years. 

Line 266 (section 3.4).  In the table 4, we have the factors associated with anemia among children in Guinea after controlling for the non-spatial random effects (M2), spatial random effects (M3) and both non spatial and spatial random effects (M4).

Point 2: The agreement between the estimated prevalence of anemia and the predicted by the models (Table 2) is outstanding. I would prefer to convert Table 2 into a Figure.

Response 2:  The results of the prevalences by region from the model by including non-spatial random effects are shown in the figure 2. 

Point 3: The labels of Figure 1 are hardly noticeable. 

Response 3: I changed the figure 

Point 4: The blue color in the map of Figure 2 is indistinguishable.

Response 4: I changed the figure 

Point 5: Impossible to read the labels of Figure 2A

Response 5: I changed the figure

Reviewer 3 Report

Barry et al. applied Bayesian spatial models to analyzing anemia among under-5 children in Guinea. My major concern is the number of regions explored geographically, isn’t there a lower administrative level? Please find below my comments for revision.

  1. The authors should be careful of making strong statements without references in the introduction.

Eg. Lines 24-26: ”The worrying aspect of anemia in children is the fact that the incidence (well on the rise) and also the mortality rate is already the highest.” What will happen in the following survey round? Is it already known?

Lines 27-28: Health specialists note that recurrent diseases in a country are both due to poverty and a cause of poverty

Line 29-30: This statement is not clear “Guinea, West African country with over 12 million inhabitants [6] is not on the fringes of the impacts of anemia.”

  1. Lines 94-95: This should be moved to the first sentence in the results. Here you are describing the sample size without mentioning the covariates
  2. Line 108: What is the difference between the natural region and the administrative region. In fact, the authors should include a map of Guinea indicating the regions (with labels)
  3. How are the covariates selected?
  4. Line 119: These are not descriptive analysis. What about frequencies and percentages? Rewrite this intro to the statistical analysis concisely.
  5. Model 1, what do you mean by logistic-normal model?
  6. Several times in the results section, the authors mentioned “bivariate (unadjusted) analysis and binary logistic regression.” What is this?
  7. What is the different between 3.2 and 3.4? There is no need to have both.
  8. Table 3 and 4 are too wordy. Write “OR (95 CrI)” and the models number only. You have already described the models in the methods.
  9. Figure 1 is not legible, and why do you need it? Isn’t that the results from model 2 (displayed in Table 4)?
  10. Malaria was mentioned briefly in the results. Why is this disease important as related to anemia? A co-infection? Discuss. The results are not well discussed, with an adequate explanation for the findings in this study. The authors should expatiate the discussion with relevant literature.
  11. Lastly, since this is a Bayesian approach, I expect to at least see the convergence plots in the appendix.

Author Response

Response to Reviewer 3 Comments and Suggestions

Point 1: The authors should be careful of making strong statements without references in the introduction

Eg. Lines 24-26: ”The worrying aspect of anemia in children is the fact that the incidence (well on the rise) and also the mortality rate is already the highest.” What will happen in the following survey round? Is it already known?

Lines 27-28: Health specialists note that recurrent diseases in a country are both due to poverty and a cause of poverty

Line 29-30: This statement is not clear “Guinea, West African country with over 12 million inhabitants [6] is not on the fringes of the impacts of anemia.”

Response 1: I added the references and I rewrote the sentence 

Lines 24-25: Anemia contributes to increased morbidity and mortality, reduced work productivity, and impaired neurological development [4–6].

Lines 27-28: Health specialists note that the level of economic development has an influence on the level of endemicity [8].

Lines 28-30: Guinea is a West African country with a population of over 12 million inhabitants [9] and is not on the fringes of the impacts of anemia.

Point 2: Lines 94-95: This should be moved to the first sentence in the results. Here you are describing the sample size without mentioning the covariates

Response 2: I moved this sentence in the results 

Lines 179-180: In total, 2609 children under 5 years who had full covariate information were used in the current analysis.

Point 3: Line 108: What is the difference between the natural region and the administrative region. In fact, the authors should include a map of Guinea indicating the regions (with labels)

Response 2: Natural regions are at a higher level than administrative regions. I included a map of Guinea with labels (Appendix B). Point 4: How are the covariates selected? Response 4: I explored in the literature.  Any variable having a significant test with anemia status is selected as a candidate for the multivariate analysis. I also added the variables not selected for the original multivariate model one at a time. This is to identify variables that, by themselves, are not significantly related to the outcome but make an important contribution in the presence of other variables. Point 5: Line 119: These are not descriptive analysis. What about frequencies and percentages? Rewrite this intro to the statistical analysis concisely Response 5:  I rewrote the sentence. 

Lines 116-120:  To this end, the following approaches of analysis are used: description of the study population (frequencies and percentages), bivariate analysis (unadjusted) , binary logistic regression modeling, and spatial analysis (models after adjusting for non-spatial random, spatial random effects) to describe heterogeneity of anemia in children in the space.

Point 6: Model 1, what do you mean by logistic-normal model?

Response 6:  It was a mistake

Model 1: Logistic regression model

Point 7: Several times in the results section, the authors mentioned “bivariate (unadjusted) analysis and binary logistic regression.” What is this?

Response 7: 

I rewrote

bivariate analysis (unadjusted) (relationship between one independent variable and binary outcome ).

logistic regression (adjusted according to the other variables within the  model).

Point 8: What is the different between 3.2 and 3.4? There is no need to have both.

Response 8: I rewrote

Line 239- 3.2 Model without spatial and non-spatial components

Section 3.2 is binary logistic model without non-spatial and spatial components

Section 3.4 is model with non-spatial and spatial components

Point 9: Table 3 and 4 are too wordy. Write “OR (95 CrI)” and the models number only. You have already described the models in the methods.

Response 9: I wrote

Point 10: Figure 1 is not legible, and why do you need it? Isn’t that the results from model 2 (displayed in Table 4)?

Response 10: I changed the figure. This figure only shows the variables that are significant

Point 11: Malaria was mentioned briefly in the results. Why is this disease important as related to anemia? A co-infection? Discuss. The results are not well discussed, with an adequate explanation for the findings in this study. The authors should expatiate the discussion with relevant literature.

Response 11: I discussed this result in the discussion part

Lines 325-333: The overall prevalence of malaria infection in children under 5 years of age was estimated at 15.14%. According to studies, it is becoming increasingly evident that human diseases are not isolated from one another [33]. Among many other factors, malaria plays a major causative role of anemia globally. The mechanisms causing anemia during malaria are extremely diverse, involving immunological factors that act differently depending on age and malaria epidemiology [34]. Almost all infants and young children have a reduced hemoglobin level in areas where malaria is prevalent [35]. In those affected by malaria, the blood is infected and the result is an abnormal drop in the number of red blood cells which compromises the rapid recovery from anemia.

Point 12: Lastly, since this is a Bayesian approach, I expect to at least see the convergence plots in the appendix

Response 12: I included the convergence plots in the appendix

Round 2

Reviewer 3 Report

The authors have responded satisfactorily to my previous comments.